# Explicitly disentangling image content from translation and rotation with spatial-VAE

**Tristan Bepler**
Massachusetts Institute of Technology
Cambridge, MA
tbepler@mit.edu

**Ellen D. Zhong**
Massachusetts Institute of Technology
Cambridge, MA
zhonge@mit.edu

**Kotaro Kelley**
New York Structural Biology Center
New York, NY
kkelley@nysbc.org

**Edward Brignole**
Massachusetts Institute of Technology
Cambridge, MA
brignole@mit.edu

**Bonnie Berger**[*]
Massachusetts Institute of Technology
Cambridge, MA
bab@mit.edu

## Abstract

Given an image dataset, we are often interested in finding data generative factors that encode semantic content independently from pose variables such as rotation and translation. However, current disentanglement approaches do not impose any specific structure on the learned latent representations. We propose a method for explicitly disentangling image rotation and translation from other unstructured latent factors in a variational autoencoder (VAE) framework. By formulating the generative model as a function of the spatial coordinate, we make the reconstruction error differentiable with respect to latent translation and rotation parameters. This formulation allows us to train a neural network to perform approximate inference on these latent variables while explicitly constraining them to only represent rotation and translation. We demonstrate that this framework, termed spatial-VAE, effectively learns latent representations that disentangle image rotation and translation from content and improves reconstruction over standard VAEs on several benchmark datasets, including applications to modeling continuous 2-D views of proteins from single particle electron microscopy and galaxies in astronomical images. [2]

## 1   Introduction

A central problem in computer vision is unsupervised learning on image datasets. Often, this takes the form of latent variable models in which we seek to encode semantic information about images into discrete (as in mixture models) or continuous (as in recent generative neural network models) vector representations. However, in many imaging domains, image content is confounded by variability from general image transformations, such as rotation and translation. In single particle electron microscopy, this emerges from particles being randomly oriented in the microscope images. In astronomy, objects

---

[*]To whom correspondences should be addressed.

[2]Source code and data are available at: `https://github.com/tbepler/spatial-VAE`

appear randomly oriented in telescope images, such as galaxies in the Sloan Digital Sky Survey. There have been significant efforts to develop rotation and translation invariant clustering methods for images in various domains [1–3]. However, learning continuous latent variable models that capture content separately from nuisance transformations remains an open problem.

We motivate the importance of this problem in particular for single particle electron microscopy (EM), where the goal is to determine the 3-D electron density of a protein from many noisy and randomly oriented 2-D projections. The first step in this process is to model the variety of 2-D views and protein conformational states. Existing methods for this step use Gaussian mixture models to group these 2-D views into distinct clusters where the orientation of each image relative to the cluster mean is inferred by maximum likelihood. This assumes that these projections arise from a discrete set of views and conformational states. However, protein conformations are continuous and may be poorly approximated with discrete clusters. Despite interest in continuous latent representations, no general methods exist.

In this work, we focus specifically on the problem of learning continuous generative factors of image datasets in the presence of random rotation and translation of the observed images. Our goal is to learn a deep generative model and corresponding latent representations that separate content from rotation and translation and to perform efficient approximate inference on these latent variables in a fully unsupervised manner. To this end, we propose a novel variational autoencoder framework, termed spatial-VAE. By parameterizing the distribution over pixel intensities at a spatial coordinate explicitly as a function of the coordinate, we make the image reconstruction term differentiable with respect to rotation and translation variables. This insight enables us to perform approximate inference on these latent variables jointly with unstructured latent variables representing the image content and train the VAE end-to-end. Unlike in previously proposed unconstrained disentangled representation learning methods (e.g. $\beta$-VAE [4]), the rotation and translation variables are structurally constrained to represent only those image transformations.

In experiments, we demonstrate the ability of spatial-VAE to disentangle image content from rotation and translation. We find that rotation and translation inference allows us to learn improved generative models when images are perturbed with random transformations. In application to single particle EM and astronomical images, spatial-VAE learns latent representations of image content and generative models of proteins and galaxies that are disentangled from confounding transformations naturally present in these datasets. Going forward, we expect this general framework to enable better, spatially-aware object models across a variety of image domains.

## 2 Methods

### 2.1 Spatial generator network

The defining component of our spatial-VAE framework is the parameterization of the deep image generative model as a function of the spatial coordinates of the image. That is, given a signal with $n$ observations indexed by $i$, we learn a single function that describes the probability of observing the signal value, $y^i$, at coordinate, $\mathbf{x}^i$, as a function of $\mathbf{x}^i$ and the unstructured latent variables, $\mathbf{z}$. For images, $\mathbf{x}^i$ is the 2-dimensional spatial coordinate of pixel $i$, but this concept generalizes to signals of arbitrary dimension. Following Kingma and Welling [5], we define this distribution, $p_g(y^i|\mathbf{x}^i, \mathbf{z})$, to be Gaussian in the case of real valued $y^i$ and Bernoulli in the case of binary $y^i$ with distribution parameters computed from $\mathbf{x}^i$ and $\mathbf{z}$ using a multilayer perceptron (MLP) with parameters $g$ (Figure 1). Under this model, the log probability of an image, $\mathbf{y}$, represented as a vector of size $n$ with corresponding spatial coordinates, $\mathbf{x}^i$, given the current MLP and unstructured latent variables, $\mathbf{z}$, is

$$\log p(\mathbf{y}|\mathbf{z}) = \sum_{i=1}^{n} \log p_g(y^i|\mathbf{x}^i, \mathbf{z}). \tag{1}$$

Although the coordinate space of an image can be represented using several different systems, we use Cartesian coordinates with the origin being the center of the image to naturally represent rotations around the image center. Therefore, to get the predicted distribution over the pixel at spatial coordinate $\mathbf{x}^i$, $\mathbf{z}$ and $\mathbf{x}^i$ are concatenated and fed as input to the MLP. We contrast our spatial generator network with the usual approach to neural image generative models in which each pixel is decoded independently, conditioned on the latent variables. In these standard models, which include

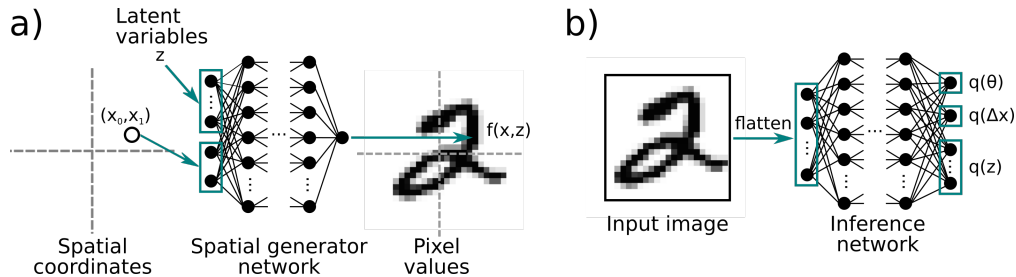

Figure 1: Diagram of the spatial-VAE framework. **a)** The generative model is a MLP mapping spatial coordinates and latent variables to the distribution parameters of the pixel intensity at that coordinate. This model is applied to each coordinate in the pixel grid to generate a complete image. Coordinate transformations are applied directly to the spatial coordinates before being decoded by the generator network. **b)** Approximate inference is performed on the rotation, translation, and unstructured latent variables using an inference network, depicted here as a MLP. We use this architecture in our experiments, but our framework generalizes to other inference network architectures.

both fully connected and transpose convolutional models, one function is learned per pixel index whereas we explicitly condition on the pixel spatial coordinate instead.

**Modeling rotation and translation**. This model structure allows us to represent rotations and translations directly as transformations of the coordinate space. A rotation by $\theta$ of the image $\mathbf{y}$ corresponds to rotating the underlying coordinates by $\theta$. Furthermore, shifting the image by some $\Delta\mathbf{x}$ corresponds to shifting the spatial coordinates by $\Delta\mathbf{x}$. Let $R(\theta) = [[cos(\theta), sin(\theta)], [-sin(\theta), cos(\theta)]]$ be the rotation matrix for angle $\theta$, then the probability of observing image $\mathbf{y}$ with rotation $\theta$ and shift $\Delta\mathbf{x}$ is given by

$$\log p_g(\mathbf{y}|\mathbf{z}, \theta, \Delta\mathbf{x}) = \sum_{i=1}^{n} \log p_g(y^i|\mathbf{x}^i R(\theta) + \Delta\mathbf{x}, \mathbf{z}). \tag{2}$$

This formulation of the conditional log probability of the image is differentiable with respect to $\theta$ and $\Delta\mathbf{x}$ which enables us to train an approximate inference network for these parameters. Furthermore, although we consider only rotation and translation in this work, this framework extends to general coordinate transformations.

## 2.2 Approximate inference of rotation, translation, and unstructured latent variables

We perform approximate inference on the unstructured latent variables, the rotation, and the translation using a neural network following the standard VAE procedure. For all parameters, we make the usual simplifying choice and represent the approximate posterior as $\log q(\mathbf{z}, \theta, \Delta\mathbf{x}|\mathbf{y}) = \log \mathcal{N}(\mathbf{z}, \theta, \Delta\mathbf{x}; \mu(\mathbf{y}), \sigma(\mathbf{y})^2 I)$. For $\mathbf{z}$, we use the usual $\mathcal{N}(0, I)$ prior. Choosing the appropriate priors on $\Delta\mathbf{x}$ and $\theta$, however, requires more consideration. We set the prior on $\Delta\mathbf{x}$ to be Gaussian with $\mu = 0$, but the standard deviation of this prior controls how tolerant our model should be to large image translations. Priors on $\theta$ are particularly tricky, due to angles being bounded and, ideally, we would like to have the option to use a uniform prior over $\theta$ between $0$ and $2\pi$ and to use an approximate posterior distribution with matching support. In particular, the wrapped normal or von Mises-Fisher distributions would be ideal. However, these distributions introduce significant extra challenges to sampling and computing the Kullback–Leibler (KL) divergence in the VAE setting [6]. Therefore, we instead model the prior and approximate posterior of $\theta$ using the Gaussian distribution ($\theta$ is wrapped when we calculate the rotation matrix). For the prior, we use mean zero and the standard deviation of this distribution can be set large to approximate a uniform prior over $\theta$. Furthermore, by observing that the KL divergence does not penalize the mean of the approximate posterior when the prior over $\theta$ is uniform, we can make the following adjustment to the standard KL divergence for $\theta$:

$$D_{\mathrm{KL}}^{\theta}(\mathbf{y}) = -\log \sigma_\theta(\mathbf{y}) + \log s_\theta + \frac{\sigma_\theta^2(\mathbf{y})}{2s_\theta^2} - 0.5 \tag{3}$$

where $\sigma_\theta(\mathbf{y})$ is given by the inference network for image $\mathbf{y}$ and $s_\theta$ is the standard deviation of the prior on $\theta$. In our experiments, we use this variant of the KL divergence for $\theta$ in cases where we would like to make no assumptions about its prior mean. In the future, this structure could be replaced with better prior and approximate posterior choices for the translation and rotation parameters.

## 2.3 Variational lower-bound for this model

Let the approximate posterior given by the inference network to the unconstrained latent variables for an image $\mathbf{y}$ be $q(\mathbf{z}|\mathbf{y}) = \mathcal{N}(\mu_\mathbf{z}(\mathbf{y}), \sigma_\mathbf{z}^2(\mathbf{y}))$, the rotation be $q(\theta|\mathbf{y}) = \mathcal{N}(\mu_\theta(\mathbf{y}), \sigma_\theta^2(\mathbf{y}))$, and the translation be $q(\Delta\mathbf{x}|\mathbf{y}) = \mathcal{N}(\mu_{\Delta\mathbf{x}}(\mathbf{y}), \sigma_{\Delta\mathbf{x}}^2(\mathbf{y}))$. For convenience, we denote these variables collectively as $\phi$ and joint approximate posterior as $q(\phi|\mathbf{y})$. The full variational lower bound for our model with inference of the rotation and translation parameters on a single image is

$$\mathbb{E}_{\phi \sim q(\phi|\mathbf{y})} [\log p_g(\mathbf{y}|\mathbf{z}, \theta, \Delta\mathbf{x})] - D_{\mathrm{KL}}(q(\phi|\mathbf{y})||p(\phi)), \tag{4}$$

where $D_{\mathrm{KL}}(q(\phi|\mathbf{y})||p(\phi)) = D_{\mathrm{KL}}(q(\mathbf{z}|\mathbf{y})||p(\mathbf{z})) + D_{\mathrm{KL}}(q(\theta|\mathbf{y})||p(\theta)) + D_{\mathrm{KL}}(q(\Delta\mathbf{x}|\mathbf{y})||p(\Delta\mathbf{x}))$, $p(\mathbf{z}) = \mathcal{N}(0, I)$, $p(\theta) = \mathcal{N}(0, s_\theta^2)$, and $p(\Delta\mathbf{x}) = \mathcal{N}(0, s_{\Delta\mathbf{x}}^2)$ with problem specific $s_\theta$ and $s_{\Delta\mathbf{x}}$. In the case that we do not wish to perform inference on $\theta$ or $\Delta\mathbf{x}$, these variables are fixed to zero when calculating the expected log probability and their KL divergence terms are ignored:

$$\mathbb{E}_{\mathbf{z} \sim q(\mathbf{z}|\mathbf{y})} [\log p_g(\mathbf{y}|\mathbf{z}, \theta = 0, \Delta\mathbf{x} = 0)] - D_{\mathrm{KL}}(q(\mathbf{z}|\mathbf{y})||p(\mathbf{z})). \tag{5}$$

When we wish to impose no prior constraints on the mean of $\theta$, the $D_{\mathrm{KL}}(q(\theta|\mathbf{y})||p(\theta))$ term is substituted for the modified KL divergence in equation 3. We estimate the expectation of the log probability with a single sample during model training.

## 3   Related Work

Extensive research on developing deep generative models of images has led to a diversity of frameworks, including Generative Adversarial Networks [7–9], Variational Autoencoders [5, 10], hybrid VAE/GANs [11], and flow-based models [12]. These models have been broadly successful for unsupervised representation learning of images and/or the synthesis of realistic images. These approaches, however, do not impose any semantic structure on their inferred latent space. In contrast, we are interested in separating latent variables into latent translation/rotation and unstructured latent variables encoding image content. Also in this category are models for unsupervised scene understanding such as AIR [13] which seeks to learn to break down scenes into constitutive objects. The individual object representations are unstructured. Instead, we seek to learn object representations and corresponding generative models that are invariant to rotation and translation by explicitly structuring the latent variables to remove these sources of variation from the object representations. In this way, our work is related to transformation invariant image clustering where it is well understood that in the presence of random rotation and translation, discrete clustering methods tend to find cluster centers that explain these transformations [1]. We extend this to learning continuous, rather than discrete, latent representations of images in the presence of random rotation and translation.

Recent literature on learning disentangled or factorized representations include $\beta$-VAE [4] and Info-GAN [14]. These approaches tackle the problem of disentangling latent variables in an unconstrained setting, whereas we explicitly detangle latent pose from image content variables by constraining their effect to transformations of input pixel coordinates. This approach has the added benefit of introducing no additional hyperparameters or optimization complexity that would be required to impose this disentanglement through modifications to the loss function. Other efforts to capture useful latent representations include constraining the manifold of latent values to be homeomorphic to some known underlying data manifold [15, 16].

The general framework of modeling images as functions mapping spatial coordinates to values has not been extensively explored in the neural network literature. The first example to our knowledge is with compositional pattern producing networks (CPPNs) [17], although their focus was on using CPPNs to model complex functions (i.e. images) as an analogy to development in nature, with the form of the image model being incidental. The only other example, to our knowledge, is given by CocoNet [18], a deep neural network which maps 2D pixel coordinates to RGB color values.

CocoNet learns an image model from single images, using the capacity of the network to memorize the image. The trained network can then be used on various low-level image processing tasks, such as image denoising and upsampling. While we use a similar coordinate-based image model, we are instead interested in learning latent generative factors of the dataset from many rotated and translated images. In the EM setting, for example, we would like to learn the distribution over protein structures from many randomly oriented images. Spatial coordinates have also been used as additional features in convolutional neural networks for supervised learning [19] and generative modeling [20]. However, the latter uses these only to augment standard convolutional decoder architectures. In contrast, we condition our generative model on the spatial coordinates to enable structured inference of pose parameters.

In EM, methods for addressing protein structural heterogeneity can broadly be characterized into ones that operate on 2D images [2] and ones that operate on 3D volumes [3, 21]. These methods assume that images arise from a discrete set of latent conformations and often require significant manual data curation to group similar conformations. To understand continuous flexibility, Dashti et al. [22] use statistical manifold embedding of prealigned protein images. More recently, multi-body refinement [23], in which electron densities of protein substructures are refined separately, has been used to model independently moving rigid domains, but requires manual definition of these components. *Our work is the first attempt to use deep generative models and approximate inference to automatically learn continuous representations of proteins from electron microscopy data.*

## 4  Results

### 4.1  Experiment setup, implementation, and training details

We represent the coordinate space of an image using Cartesian coordinates with the origin at the image center. The coordinates are normalized such that the left-most and bottom-most pixels occur at -1 and the right-most and top-most pixels occur at +1 along each axis. The generator network is implemented as an MLP with tanh activations. The input dimension is 2 + the dimensions of the unstructured latent variables (Z-D). The output is either a single output probability, in the case of binary valued pixels, or a mean and standard deviation in the case of real valued pixels. In the following experiments, the binary valued pixel log probability is used for MNIST and the galaxy zoo dataset and the real valued pixel log probability is used for the EM datasets. In all cases, the inference network uses the same MLP architecture as the generator, except that the inputs are flattened images and the outputs are mean and standard deviations of the latent variables. For comparison, we define a vanilla VAE using a standard MLP generator net-

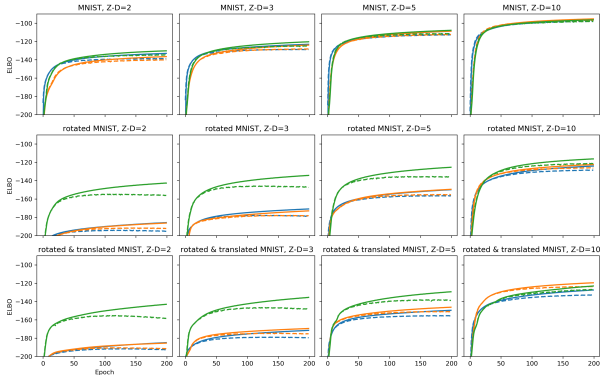

Figure 2: Comparison of VAEs in terms of the ELBO for varying dimensions of the unstructured latent variables (Z-D) on MNIST and transformed MNIST datasets. (Green) spatial-VAE, (orange) spatial-VAE trained with fixed $\theta = 0$ and $\Delta \mathbf{x} = 0$, (blue) standard VAE baseline. Spatial-VAE achieves better ELBO on both transformed datasets with more pronounced benefit when the dimension of the latent variables is small. Remarkably, the spatial-VAE even gives some benefit on the plain MNIST dataset with small $\mathbf{z}$ dimension, likely due to slight variation in digit slant and positioning.

work in which the Z-D latent variables are mapped directly to the distribution parameters for each pixel (i.e. the model outputs an $n$-dimensional vector for binary data and $2n$-dimensional vector for real valued data where $n$ is the number of pixels in the image). Parameters are fit to maximize the evidence lower bound (ELBO). All models are trained using ADAM [24] with a learning rate of 0.0001 and minibatch size of 100. Models were implemented using PyTorch [25].

## 4.2 Spatial-VAE improves reconstruction when images are transformed

We first ask whether our spatial-VAE model can succesfully reconstruct image content when images have been transformed through random rotation and translation. To this end, we generate two randomly transformed variants of MNIST. In the first, which we refer to as "rotated MNIST", each MNIST digit is randomly rotated by an angle sampled from $\mathcal{N}(0, \frac{\pi^2}{16})$ and randomly translated by a small amount, $\mathcal{N}(0, 1.4^2)$. We then generate a second, harder dataset, where the MNIST digits are randomly rotated by the same amount but we apply much greater translation, sampling the horizontal and vertical shift from $\mathcal{N}(0, 14^2)$. We refer to this as the rotated and translated MNIST dataset.

We train spatial-VAE models with rotation and translation inference on these datasets and the original MNIST dataset. The prior on $\theta$ is set to $\mathcal{N}(0, \frac{\pi^2}{64})$ in the case of regular MNIST where there is very little rotation of the digits and $\mathcal{N}(0, \frac{\pi^2}{16})$ for the two rotated MNIST datasets. We infer translations with the prior set to $\mathcal{N}(0, 1.4^2)$ on both transformed MNIST datasets. The encoder and decoder architectures are both 2-layer MLPs with 500 hidden units each and tanh activations. The encoder takes a 28*28 dimension input, encoding a flattened image, and outputs the mean and standard deviation of the approximate posterior distribution over each of the unconstrained latent variables, the rotation, and the translation. The decoder network takes the unconstrained latent variables and the rotated and translated x and y coordinates of each pixel and returns the reconstructed pixel value at that coordinate.

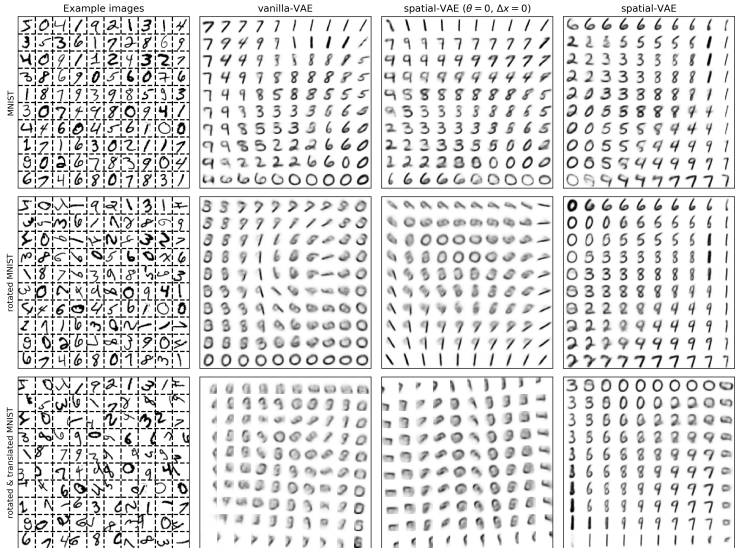

Figure 3: Visualization of the learned data manifolds of MNIST and transformed MNIST for models with 2-D unconstrained latent variables. We plot $p_g(\mathbf{y}|\mathbf{z}, \theta = 0, \Delta\mathbf{x} = 0)$ for values of the latent variable $\mathbf{z}$ equally spaced through the inverse CDF. The vanilla VAE and spatial-VAE with $\theta$ and $\Delta\mathbf{x}$ fixed to 0 are forced to model digit orientation only with $\mathbf{z}$, whereas the full spatial-VAE model only uses $\mathbf{z}$ to capture digit style.

We compare the performance of the spatial-VAE against two baselines. The first is a standard (vanilla) autoencoder in which each pixel value is decoded directly from the unconstrained latent variables also using a 2-layer MLP with 500 hidden units and tanh activations (this model takes as input Z-D latent variables and outputs a 28*28 dimension vector giving the reconstructed pixel values). The second is a spatial-VAE with identical architecture to above, but without rotation and translation inference (i.e. $\theta = 0$ and $\Delta\mathbf{x} = 0$ for all images). Our spatial-VAE model outperforms both baselines in terms of maximizing the ELBO of the test set images on all three datasets across a variety of sizes of the unconstrained latent variables (Figure 2). As expected, spatial-VAE provides an enormous improvement on both transformed MNIST datasets when there are few unconstrained latent variables. Even when the standard VAE is given additional unstructured latent variables to account for rotation and translation, spatial-VAE still achieves identical performance (Appendix Figure 1). As the dimension of $\mathbf{z}$ increases, the models lacking rotation and translation inference are able to account

for this variability within **z** so the size of the improvement decreases. Remarkably, spatial-VAE even offers some improvement on untransformed MNIST when the dimension of **z** is small, perhaps because there is some small variability in digit orientation. This is evident in the manifold of digits learned by these models (Figure 3, where we see that the spatial-VAE model encodes digit shape but not rotation and translation in the unconstrained latent variables). Spatial-VAE learns to generate recognizable digits even from the transformed MNIST images whereas the baseline models do not. Although we set the rotation and translation priors to match the true distribution above, we observe that spatial-VAE is not sensitive to this setting and achieves the same results with wide priors on these parameters (Appendix Figures 2 & 3).

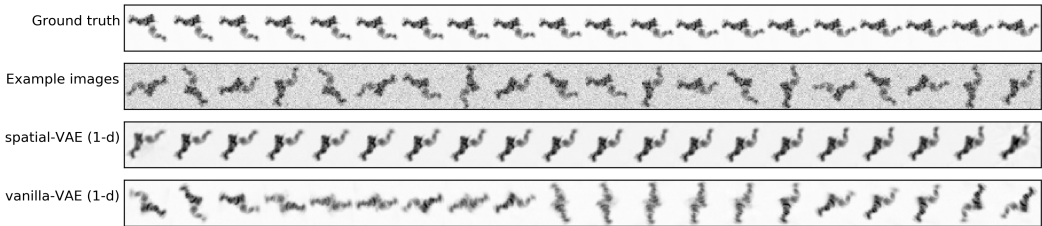

Figure 4: Visualization of the ground truth conformations of 5HDB (top), simulated particle samples with random rotation and added noise (second from top), visualization of the 1-D data manifold learned by spatial-VAE (second from bottom), and visualization of the 1-D data manifold learned by the vanilla VAE (bottom). The spatial-VAE model captures the protein's conformation in the unconstrained latent variable separately from protein orientation.

### 4.3 Spatial-VAE recovers true variation in image content

We next ask if the spatial-VAE can recover true semantic variability in images that have been observed with random transformations. Specifically, we consider the ability of our spatial-VAE framework to capture protein conformation separately from in-plane rotation in simulated single particle cryo-EM data. To this end, we generated 20,000 simulated projections of integrin $\alpha$-IIb in complex with integrin $\beta$-3 (5HDB) [26]) with varying conformations given by a single degree of freedom (see appendix for details). These images are 40 by 40 pixels and were randomly split into 16,000 train and 4,000 test images. We then fit a spatial-VAE with a 1-D unconstrained latent variable and rotation inference. This is compared with vanilla VAEs trained with one- and two-dimensional latent variables. All models are 2-layer MLPs with 500 hidden units in each layer and tanh activations. Models were trained for 500 epochs. Because the simulated particles have a uniform prior on the orientation angle, we maximize the variational lower bound with the KL divergence variant presented in equation 3 with the prior on $\theta$ set to have $\sigma = \pi$. In order to avoid bad local optima that could arise early on during spatial-VAE training, we freeze the unconstrained latent variable to zero for the first two training epochs. Furthermore, we apply data augmentation to the inference network by randomly rotating images by $\gamma$ and then removing $\gamma$ from the predicted rotation angle, where $\gamma$ is randomly sampled from $[0, 2\pi]$ for each image at each iteration.

Our spatial-VAE dramatically outperforms the 1-D vanilla VAE and slightly outperforms the 2-D vanilla VAE in terms of ELBO despite being constrained to only represent rotation with its second latent variable (Appendix Figure 4). Furthermore, in visualizations of the learned data manifold (Figure 4), we see that spatial-VAE correctly reproduces the ground truth protein conformations with orientation removed. The vanilla VAE, on the other hand, does not

| Model | Variable | Conformation | Rotation |
|---|---|---|---|
| vanilla-VAE (1-d) | $z_1$ | 0.00 | 0.18 |
| vanilla-VAE (2-d) | $z_1$ | 0.09 | 0.02 |
| vanilla-VAE (2-d) | $z_2$ | 0.07 | 0.04 |
| spatial-VAE | $z_1$ | **0.95** | 0.01 |
| spatial-VAE | $\theta$ | 0.01 | **0.92** |

Table 1: Correlation coefficients of the inferred latent variables with the ground truth factors in the 5HDB dataset.

separate rotation and conformation in the latent space (Appendix Figure 5). We confirm this finding quantitatively by measuring the correlation between the latent variables inferred by these models and the ground truth rotation and conformation factors (Table 1). For the conformation factor we calculate

Pearson correlation and for the rotation factor we calculate the circular correlation measure [27]. The latent space learned by spatial-VAE correlates strongly with the ground truth conformation whereas the latent spaces learned by the standard VAEs do not. The same is true for the inferred rotation.

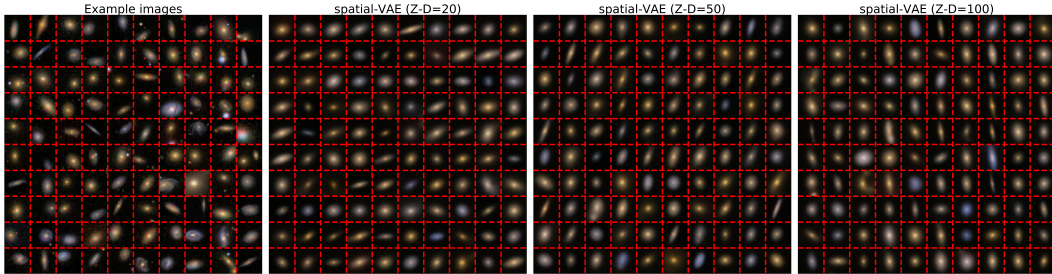

Figure 5: Visualization of samples from the galaxy zoo spatial-VAEs. (**Left**) Random examples from the training images showing the diversity of galaxy shapes, sizes, and colors. These differences are further confounded by random rotation and position of galaxies within the image frame. (**Left-middle**, **right-middle**, **right**) Samples from spatial-VAE models with 20-, 50-, and 100-D unconstrained latent variables. $\mathbf{z}$ is first sampled from $\mathcal{N}(0,1)$, then $p_g(\mathbf{x}|\mathbf{z}, \theta = 0, \Delta\mathbf{x} = 0)$ is plotted for each RGB value. These samples demonstrate the diversity of shapes, sizes, and colors while being invariant to rotation and translation. Best viewed zoomed in.

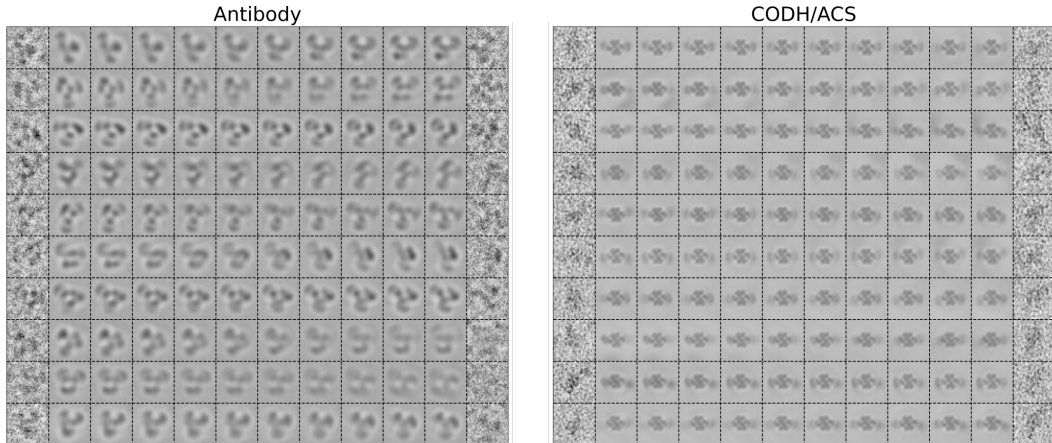

Figure 6: Visualization of interpolation between observed antibody conformations (**Left**) and observed CODH/ACS conformations (**Right**) in the latent space of the 20-D spatial-VAEs trained on each dataset. We plot the mean of the pixel distribution given by the generator network for $\mathbf{z}$ interpolated along equally sized steps between the mean of the approximate posterior distributions given by the inference network to randomly selected test image pairs. We show the sampled images in the far left and far right columns respectively. Note that the visualizations are generated with $\theta = 0$ and $\Delta\mathbf{x} = 0$, which removes the orientation from the observed images.

### 4.4 Learning transformation-invariant protein and galaxy models

We demonstrate that our spatial-VAE can capture conformational variability separately from rotation and translation in astronomical images from the galaxy zoo dataset and in real single particle electron-microscopy images.

**Galaxy zoo.** The galaxy zoo dataset contains 61,578 training color images of galaxies from the Sloan Digital Sky Survey. We crop each image with random translation and downsample to 64x64 pixels following common practice [28]. We train spatial-VAEs with 20, 50, and 100 dimension unconstrained latent variables for 300 epochs following the description above except that our inference network has 5,000 units in each hidden layer. Furthermore, because these are RGB images, our generator network outputs three values given the spatial coordinate rather than one. We use the

KL divergence variant in eq. 3 and set the translation prior standard deviation to 8 pixels. We find that spatial-VAE captures galaxy size, shape, and color independently from rotation and translation (Figure 5).

**Single particle electron-microscopy.** We also train spatial-VAEs on two negative stain EM datasets. The first is a dataset containing the StrepMAB-Classic antibody and the second contains the CODH/ACS protein complex (see appendix for details). We split the antibody dataset into 10,821 training and 2,705 testing images and the CODH/ACS dataset into 11,473 training and 2,868 testing images. Each image is 40 by 40 pixels. Models are 2-layer MLPs with 1,000 hidden units in each layer and tanh activations and are trained for 1,000 epochs. Again, we use the KL-divergence variant in equation 3 with prior $\sigma = \pi$. We also perform inference on $\Delta \mathbf{x}$ with a $\mathcal{N}(0, 4)$ pixel prior and apply random rotation augmentation to the inference network training.

Consistent with other work in VAEs and our above results, we do not observe that adding additional latent variables causes overfitting of spatial-VAE (Appendix Figure 6). Furthermore, spatial-VAE recovers continuous conformations of the proteins in these datasets. Figure 6 visualizes interpolation between images using the spatial-VAE model trained with 20-D $\mathbf{z}$. EM images have low signal-to-noise ratios, but the random orientations of the proteins can be seen in the real images in the left and right columns. In the CODH/ACS dataset, spatial-VAE learns a variety of configurations of the "arms" of the complex. We note that spatial-VAE models also capture structure in the image background (Appendix Figures 7 & 8) which can be mitigated by constraining the mean of the pixel value distribution given by the generator network to be non-negative.

## 5 Conclusion

We have presented spatial-VAE, a method for learning latent image representations disentangled from rotation and translation. We showed that formulating the image generative model as a function of the spatial coordinates not only enables efficient inference of the pose parameters but that this framework leads to improved image modeling. Furthermore, our general approach is highly extensible. The spatial generator network can operate on signals of any dimension, suggesting that this could be a promising approach to 3-D object modeling, although this application will require additional work in high dimensional pose inference and image formation processes. As a second direction, decoupling rotation, translation, and semantic content in the inference network could lead to improvements in the inference process. We are hopeful that these ideas will enable new directions in generative models of images and objects.

## Acknowledgements

This work was supported by NIH R01-GM081871.

We would like to thank Bridget Carragher and Clint Potter at NYSBC for their support in providing the antibody dataset. The NYSBC portion of this work was supported by Simons Foundation SF349247, NYSTAR, and NIH GM103310 with additional support from Agouron Institute F00316 and NIH OD019994-01.

We would also like to thank the laboratory of HHMI investigator Catherine L. Drennan, MIT, for providing the CODH/ACS dataset that was collected with support from the National Institutes of Health (R35 GM126982).

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
