[Supplementary Material]

# 1 Appendix

## 1.1 Simulated EM Data

A crystal structure of integrin -IIb in complex with integrin -3 (5HDB, Lin et al. [1]) was used to simulate in-plane rotation and rigid body flexibility. Rigid body hinge motion was simulated by manually rotating integrin -3 around its binding interface with integrin -IIb over 20 steps in 2 increments. Density maps were generated in Chimera [2] with the molmap command, randomly rotated in-plane, and projected along the z axis.

## 1.2 Sample preparation and imaging

**Antibody.** StrepMAB-Classic antibody was obtained from IBA Lifesciences and used without further purification. Both protein samples were diluted to 15 ng/ul with buffer containing 200 mM NaCl and 20 mM HEPES pH 7.4 and stained with 2% uranyl formate on continuous carbon grids. Microscopy was performed with an FEI Tecnai 12 BioTwin operated at 120 kV and TVIPS F416 CMOS detector used with a 100 m C2 aperture, 100 m objective aperture and calibrated pixel size of 2.46Å. Images were collected using Leginon with a nominal defocus of -3 um [3]. Particles were picked automatically from micrographs using the Appion DoG Picker [4] and extracted into a single particle stack.

**CODH/ACS.** Preparation, imaging, and analysis of CODH/ACS will be described in detail in a separate publication. Briefly, 5 L of 17 ng/L Moorela thermoacetica CODH/ACS in storage buffer (50 mM Tris, pH 7.6, 100 mM NaCl) was applied to carbon-coated EM grids and negative stained with 2% uranyl acetate. 131 images of the specimen at 62,000x magnification corresponding to pixel size of 1.79Åwere acquired as 1 sec (23 e - /Å$^2$) exposures with a Gatan US4000 CCD camera on a FEI Tecnai F20 electron microscope operated at 200 kV using SerialEM. 19,393 particle images were extracted using EMAN2 libraries [5] and then aligned and classified using Iterative Stable Alignment and Clustering to create 44 stable class averages comprising 14,341 particles. A subset of 14,341 particles from the class averages of well-defined intact CODH/ACS was provided for this analysis.

# References

[1] Fu-Yang Lin, Jianghai Zhu, Edward T Eng, Nathan E Hudson, and Timothy A Springer. $\beta$-subunit binding is sufficient for ligands to open the integrin $\alpha$iib$\beta$3 headpiece. *Journal of Biological Chemistry*, 291(9):4537–4546, 2016.

[2] Eric F Pettersen, Thomas D Goddard, Conrad C Huang, Gregory S Couch, Daniel M Greenblatt, Elaine C Meng, and Thomas E Ferrin. Ucsf chimera—a visualization system for exploratory research and analysis. *Journal of computational chemistry*, 25(13):1605–1612, 2004.

[3] Christian Suloway, James Pulokas, Denis Fellmann, Anchi Cheng, Francisco Guerra, Joel Quispe, Scott Stagg, Clinton S Potter, and Bridget Carragher. Automated molecular microscopy: the new leginon system. *Journal of structural biology*, 151(1):41–60, 2005.

[4] NR Voss, CK Yoshioka, M Radermacher, CS Potter, and B Carragher. Dog picker and tiltpicker: software tools to facilitate particle selection in single particle electron microscopy. *Journal of structural biology*, 166(2):205–213, 2009.

[5] Guang Tang, Liwei Peng, Philip R Baldwin, Deepinder S Mann, Wen Jiang, Ian Rees, and Steven J Ludtke. Eman2: an extensible image processing suite for electron microscopy. *Journal of structural biology*, 157(1):38–46, 2007.

 # 2 Appendix Figures

Figure 1: Reconstruction error on rotated MNIST (top row) and rotated and translated MNIST (bottom row) datasets for spatial-VAE models trained with 2-d latent variables using the original prior ($\sigma_\theta = \frac{\pi}{4}$, $\sigma_{\Delta x} = 1.4$) (left column) and a wide prior ($\sigma_\theta = \pi$, $\sigma_{\Delta x} = 14$) (right column). The models trained with the wide prior achieve exactly the same reconstruction error as the models trained with prior correctly matched to the data distribution. This indicates that the spatial-VAE model is insensitive to misspecification of the translation and rotation priors.

Figure 2: Manifolds generated from spatial-VAE models with 2-d latent variables trained on rotated MNIST and rotated and translated MNIST datasets using the original prior ($\sigma_\theta = \frac{\pi}{4}$, $\sigma_{\Delta x} = 1.4$) and a wide prior ($\sigma_\theta = \pi$, $\sigma_{\Delta x} = 14$). The models trained with the wide prior learn comparable digit manifolds to those learned by the models with correctly matched priors.

Figure 3: Comparison of generative models on the simulated 5HDB particles. The spatial-VAE with 1-D latent variable and rotation inference outperforms 1-D and 2-D latent variable vanilla VAEs in terms of the ELBO on the heldout data.

Figure 4: Visualization of the data manifold learned by the 2-d vanilla VAE on the 5HDB dataset. The vanilla VAE does not separate rotation from conformation in the latent space.

Figure 5: Comparison of spatial-VAE models trained with varying dimension of the unconstrained latent variables on the antibody and CODH/ACS datasets in terms of ELBO. Larger dimension unconstrained latent variables improve the variational lower bound and do not seem to cause overfitting.

Figure 6: Visualizations from the generative networks trained on the antibody dataset. (**Top left**) the data manifold of the 2-d latent variable model. (**Top right**) samples from the 5-d latent variable model. (**Bottom left**) samples from the 10-d latent variable model. (**Bottom right**) samples from the 20-d latent variable model.

Figure 7: Visualizations from the generative networks trained on the CODH+ACS dataset. **(Top left)** the data manifold of the 2-d latent variable model. **(Top right)** samples from the 5-d latent variable model. **(Bottom left)** samples from the 10-d latent variable model. **(Bottom right)** samples from the 20-d latent variable model.