[Reviews · NeurIPS 2019]

Reviewer 1



Originality: The formulation of the variational encoder as an explicit function of spatial coordinates is novel. Application to electron microscopy data is also novel Quality: The work appears technically sound from what I could tell. This work is a bit out of my area of expertise. Clarity: The writing is clear with sufficient background Significance: The model is novel and adds to the repertoire of generative models for images with explicit disentangled continuous representations. It opens up the door to many applications and further improvements.

Reviewer 2



1. Mentioning some solid motivations and practical usage of rotation and translation invariant latent variables would be helpful. 2. Not very clear where the MLP in Figure 1 is inserted within the VAE? A diagram of the overall architecture would be useful. 3. The method description is a bit too high level and sparse. Would be great to have some key implementation details. 4. Is Gaussian approximation appropriate for theta (i.e. angle)? 5. For the MNIST experiments, were three spatial VAEs built, one for each (transformed) MNIST dataset? How sensitive are the results to the choice on prior values, e.g. what happens if use the same value for prior on theta? Also, is setting D to 2, 3, and 5 reasonable? Based on just ELBO, seems like using higher D will do the trick. Hence, need to show usefulness of rotation and translation invariant latent variables. 6. All the nicely aligned images in Figures 4 to 6 are great, but what exactly can we extract from these? Some quantitative evaluation, and again demonstration of usefulness are highly needed. ------ My comments have been satisfactorily addressed especially the point regarding quantitative evaluation. I have thus raised the score.

Reviewer 3



The stated contribution of the paper is an unsupervised method on image data (potentially other signals on a 2D plane) for learning latent factors disentangled from orientation-preserving isometries on the plane (compositions of rotation/translation). The authors take the approach of using a single function to model decoder/likelihood functions per pixel parameterized by a global latent state and pixel position (represented by Cartesian coordinates), as opposed to the typical approach with separate functions per pixel. They disentangle orientation-preserving isometries by adding latent variables parameterizing rotation/translation which are applied to the pixel position before evaluation of the decoder/likelihood function. These latent variables are given Gaussian priors and approximate posteriors. They demonstrate improvements in ELBO for spatial VAE vs vanilla VAE with same dimension of global latent variables (excluding spatial-VAEs additional latent dimensions for rotation/translation -- we discuss this further below). They show on a few datasets how image orientation is disentangled in generated images when rotation/translation variables are fixed. Overall, we liked the ideas and the work presented. We admit were somewhat surprised similar solutions haven't been tried in this area but the authors list related work so we conclude the approach is indeed novel and appear effective. We particularly liked Fig3 for improved interpretability. We did have several issues listed below: (1) The exposition of the model in section 2.1 should be edited for improved clarity. In particular, the contrast between vanilla VAE and spatial VAE could be better highlighted, and the notation to represent different parts of the problem should be explained prior to using them. In particular: (a) The second sentence of this section (lines 60-68) could be broken up for clarity. (b) Image coordinates should be defined previously before using them as part of inline sentence description of a function. It is not immediately clear that there are $n$ pixels indexed by $i$ during this sentence. (c) It is unclear from the current description that $x^i$ represents a point on the plane rather than a scalar value. Some possible ways to address would be explicitly parameterizing at first as an ordered pair or typesetting in bold. (d) Figure 1 could separate the components of the first layer coming from the latent variables vs spatial coordinates to show how they go into the MLP. (2) In Figure 2 (a) Caption is missing description distinguishing solid vs dotted lines (b) For consideration of ELBO, we believe it would be more meaningful to consider the effective total number of latent variables in the model rather than the number of “unconstrained” latent variables. For example, we think that spatial VAE with Z-D=2 (first column of current figure) would best be compared to vanilla/fixed spatial VAE with Z-D=5 (third column of current figure) to account for the 3 latent factors representing rotation/translation. We believe that the discussion/figure should be updated to consider these differences. (3) In section 2.2, lines 102-104, it is stated that the prior and approximate posterior of $\theta$ is modeled using the Gaussian distribution with zero mean. This conflicts with the definition of the approximate posterior in line 115. We suspect that the intention is for only the prior to be constrained to be mean zero -- this can easily be fixed by removing the statement about the mean from that sentence and subsequently constraining the prior alone in the following sentence. (4) While we are not vision experts it seems the work presented here suffers from several limitations compared to approaches taken in that field. First the orientation/translations are global while in many vision problems several objects, each with it's own transformations, are involved. Second, there is a difference between transformations of the object and transformations of the camera/viewer pitch/yaw/roll. The problems discussed here are much simpler. Worth mentioning/discussing.

[Author Response · NeurIPS 2019]

We thank the reviewers for their positive comments and helpful feedback. Following these suggestions, we have made improvements to the clarity of the methods and experiments. We will also include an updated Figure 1 illustrating the full VAE framework in the next draft of the manuscript. We respond to the specific comments of the reviewers below.

**Reviewer #1** We thank the reviewer for their positive comments and interest in our work.

**Reviewer #2** We appreciate the reviewer's feedback and have made several changes to the manuscript to address the reviewer's concerns. We have improved the methods to provide additional intuition and implementation details.

1. Any imaging domain in which rotation and translation occur as nuisance variables will benefit from this approach. Single particle electron microscopy is a high impact application domain with exactly this problem. Understanding continuous variability in proteins imaged with EM is a pressing problem for which our method provides the first solution framework.

2. The MLP in figure 1 is specifically the generative network component of the VAE. The inference network is structured in the standard manner for fully connected inference networks. We will change Figure 1 to illustrate the full VAE framework and better illustrate the generative network.

3. We have updated the methods section to give more intuition and improve the description of the method. We will also release the source code with the camera-ready version of the manuscript, which will provide all implementation details.

4. Empirically, the Gaussian approximation works well, but we plan to explore other distributions in the future.

5. One network was trained for each dataset. We now clarify this in the text. These results are robust to the choice of prior values. We show in Appendix Figures 1 & 2 that spatial-VAEs trained with wide priors on these parameters learn the same manifold over digits and reach the same reconstruction error as the models with correctly matched priors. For the dimension of the unstructured latent variables, these settings represent a reasonable trade off between interpretability/compression and representation power. It is not surprising that with large $z$ dimension the ELBOs become similar, as eventually there is enough capacity in $z$ to represent both the content and the rotation and translation. However, the standard VAEs do not disentangle pose from content.

6. The purpose of these figures is to illustrate that the spatial-VAE successfully learns disentangled representations on real datasets. In Figure 4, the spatial-VAE, but not the standard VAE, recovers the ground truth variability in the dataset. As additional quantitative support for this claim, we now report the ELBOs for each model in the main text (was Appendix Figure 3) and also include a quantitative assessment of the ability of the spatial-VAE to recover the ground truth variability. We report the

| Model | Variable | Conformation | Rotation |
|---|---|---|---|
| vanilla-VAE [Z-D=1] | $z_1$ | 0.00 | 0.18 |
| vanilla-VAE [Z-D=2] | $z_1$ | 0.09 | 0.02 |
| vanilla-VAE [Z-D=2] | $z_2$ | 0.07 | 0.04 |
| spatial-VAE | $z_1$ | **0.95** | 0.01 |
| spatial-VAE | $\theta$ | 0.01 | **0.92** |

Table 1: Correlation coefficients of the inferred latent variables with the ground truth factors in the 5HDB dataset.

correlation coefficient of the mean of the approximate posterior for each latent variable with the known conformation and rotations of each image (Table 1). The latent variables learned by the standard VAEs do not separate into the ground truth conformation and rotation variables whereas the spatial-VAE latent variables correlated well with these features.

**Reviewer #3** We thank the reviewer for their helpful comments. We will clarify the method description in the final draft and will provide a comparison with the same effective total number of latent variables in the fixed/vanilla VAEs.

1. Section 2.1 and Figure 1 will be revised as suggested.

2. In Figure 2, the solid lines are training set ELBOs and the dashed lines are test set ELBOs. We now include this in the caption. Furthermore, we will include a comparison with the fixed/vanilla VAEs with the same effective total number of latent variables. For the transformed MNIST datasets, the spatial-VAEs with rotation/translation inference still outperform the standard VAEs even with the additional latent variables. We will update the discussion accordingly.

3. The reviewer is correct. Only the prior is defined to have mean zero. We have corrected this error in the text.

4. It is true that this work is limited to modeling global transformations and thus single objects. We think that extending this idea to handle multiple objects is an exciting future direction, which we now mention in the conclusion. Regarding object vs. camera transformations, generally speaking, object transformations and camera transformations are exact inverses. However, there are some interesting effects that can occur with light photography that are related to the angle of view and distance from the camera to the object (e.g. foreshortening, depth of field, etc.). Adaptation of this framework to explicitly handle these kinds of effects would also be an interesting future direction.

[Meta-Review · NeurIPS 2019]

The spatial VAE idea for images is novel and the execution of the paper was solid. The reviewers found the rebuttal convincing and have a clear consensus on accepting the paper.